# Advances of Anti-Caries Nanomaterials

**DOI:** 10.3390/molecules25215047

**Published:** 2020-10-30

**Authors:** Hui Chen, Lisha Gu, Binyou Liao, Xuedong Zhou, Lei Cheng, Biao Ren

**Affiliations:** 1State Key Laboratory of Oral Diseases & National Clinical Research Center for Oral Diseases, Sichuan University, Chengdu 610041, China; chenhuidentist@163.com (H.C.); liaobinyou@126.com (B.L.); zhouxd@scu.edu.cn (X.Z.); 2Department of Operative Dentistry and Endodontics, Sichuan University, Chengdu 610041, China; 3Department of Operative Dentistry and Endodontics, Guanghua School of Stomatology and Guangdong Provincial Key Laboratory of Stomatology, Sun Yat-sen University, Guangzhou 510055, China; gulisha@mail.sysu.edu.cn

**Keywords:** anti-caries, nanotechnology, nanomaterials, antibacterial, remineralization

## Abstract

Caries is the most common and extensive oral chronic disease. Due to the lack of anti-caries properties, traditional caries filling materials can easily cause secondary caries and lead to treatment failure. Nanomaterials can interfere with the bacteria metabolism, inhibit the formation of biofilm, reduce demineralization, and promote remineralization, which is expected to be an effective strategy for caries management. The nanotechnology in anti-caries materials, especially nano-adhesive and nano-composite resin, has developed fast in recent years. In this review, the antibacterial nanomaterials, remineralization nanomaterials, and nano-drug delivery systems are reviewed. We are aimed to provide a theoretical basis for the future development of anti-caries nanomaterials.

## 1. Introduction

Caries is one of the most important diseases, which has seriously threatened human oral health, and even the whole body, for a long time [1]. According to statistics associated with 328 major diseases from “The Lancet-Global Burden of Disease Study 2016, GBD”, the prevalence of permanent teeth caries was the highest [2]. Based on modern theory in caries etiology, the imbalance of oral flora could result in acid accumulation and lead to tooth demineralization, inducing the caries formation. At present, caries treatment is still based on filling restoration. However, clinical studies showed that dental filling and bonding materials’ lack of anti-caries properties could result in a high incidence of secondary caries. The five-year failure rate in filling restoration is up to 50%. The replacement of restorations caused by filling failure leads to many public health resources waste. Therefore, the development of new anti-caries materials is a hot spot in the field of caries prevention and treatment.

Given the current problems and challenges in caries prevention and treatment, anti-caries nanomaterials have become a breakthrough in caries research. Nanotechnology was first proposed by Richard Feynman, which is a technology that studied the properties and applications of materials in the range of 1–100 nm. Nanomaterials are superior to traditional materials due to their physical/chemical advantages, such as volume effect, surface effect, quantum size, quantum tunnel, and dielectric confinement [3]. At present, the nanoparticles used include nanorings, nanopores, nanotubes, carbon, nanocapsules, nanospheres, and dendrimers [4]. Nanoparticles can be combined with polymers or coated on different biomaterials. The smaller the diameter of nanoparticles, the larger the specific surface area, and the stronger mechanical properties and antibacterial effect [5]. They can be used as the carrier of antibacterial drugs since nanoparticles have the characteristics of targeting antibacterial with the least side effects on the host. At present, a variety of antibacterial mechanisms of nanoparticles have been proposed, including metal ion release [6], oxidative stress [7], and non-oxidative mechanisms [8]. It is widely believed that the positively charged nanoparticles are attracted to the negatively charged cell membrane of bacteria by static electricity, which changes the permeability of the cell wall, leading to cell membrane rupture and organelle leakage.

In recent years, nanotechnology has shown great application prospects in the development of anti-caries materials, especially in nano-adhesive and nano-composite resin. In this paper, the antibacterial nanomaterials, remineralization nanomaterials, nano-drug delivery systems will be reviewed. We are aimed to provide the theoretical basis for the future development of anti-caries nanomaterials.

## 2. Metal Nanoparticles Used in Caries Infections

### 2.1. Silver Nanoparticle (NAg)

Silver has a broad spectrum of antibacterial properties, which can inactivate enzymes and prevent DNA replication in bacteria. NAg further increases surface area ratio, making silver particles smaller and antibacterial effects better. When NAg was added to the adhesives, its antibacterial property increased significantly, but whether it had any effect on the bonding strength is still controversial. Some scholars have found that the antibacterial effect of NAg increased in a dose-dependent manner from 0.05%~0.1% with no effect on bonding strength or color [9]. However, ElkassasDW et al. [10] indicated that self-etching adhesive containing 0.05%~0.1% NAg can affect the bonding strength at pH 2.7. Cheng et.al [11] demonstrated that 0%~0.175% of NAg composite resin can significantly reduce biofilm growth and metabolic activity, in which antibacterial activity increased in a dose-dependent manner. Moreover, when the mass fraction of NAg is 0%~0.088%, the flexure strength and elastic modulus of modified composite resin are not significantly different from the control group. However, the flexural strength of modified composite resin decreased significantly when the nano-silver mass fraction was higher than 0.175% [12]. Mohadese Azarsina examined the antibacterial activity of NAg added in Z250 composite. The results showed that NAg could significantly inhibit the growth of *Streptococcus mutans* and *Lactobacillus* on the composite surface [13]. NAg can be released slowly from the materials and has good biocompatibility in a certain concentration range [14].

In recent years, NAg has been added to the resin in cooperation with other antibacterial agents, which has a good antibacterial effect against caries-related bacteria (Table 1). In other research, Shenggui Chen et al. studied the antibacterial effect of polymethyl methacrylate (PMMA)-cellulose nanocrystals (CNCs)-NAg modified resin against *Staphylococcus aureus* and *Escherichia coli*. The authors concluded that PMMA-CNCs-NAg modified resin exhibited excellent mechanical properties, desirable biocompatibility, and excellent antibacterial activities [14]. Fang Li et al. found that the novel antibacterial adhesives containing quaternary ammonium dimethacrylate (QADM) and NAg are promising anti-caries nanomaterials. QADM and NAg can be used as antibacterial agents on the resin surface with good bacteriostatic effect [15]. In vitro artificial enamel caries model, reduced graphene oxide-silver nanoparticles (rGO/NAg) composite showed shallower lesion depth and less mineral loss compared with the control groups [16]. In the artificial dentine caries model, Irene et al. found that polyethylene glycol-coated silver nanoparticles (PEG-NAg) can remineralize dentine caries and inhibit collagen degradation without causing significant tooth staining [17]. In the rat caries model, NAg coated orthodontic brackets inhibited *Streptococcus mutans* for 30 days and reduced caries on the smooth surfaces [18].

### 2.2. Nano-Zinc (NZn) and Nano-Zincoxide (NZnO)

NZn has a wide antibacterial spectrum, which antibacterial ability mainly comes from the quantum size effect of dissolving and releasing zinc nanoparticles. Matrix metalloproteinases (MMPs) can be activated in both total-etch and self-etch adhesives, which induce degradation of resin and dentin matrix, shortening the service life of adhesives. NZn can reduce the expression of MMPs and prolong the lifespan of adhesive. It has been reported that the addition of Zn^2+^ to total-etch adhesive can inhibit MMPs activity, reduce the decomposition of dentin collagen bundle, and protect mineral crystal formation at the eosin–tooth interface, improving the nano-mechanical properties [19,20]. Compared with micron ZnO, NZnO has higher surface potential energy and can release more zinc ions to kill bacteria. Besides, some scholars believed that NZnO could also activate the photocatalytic antibacterial mechanism and produce a large number of free radicals to interact with bacteria [21]. Tavasoli et al. [22] studied 0–5% NZnO composite resin and found that 1% NZnO had no obvious effect on the mechanical properties of the composite resin. With the increase of mass fraction (0–5%), the number of *Streptococcus mutans* was decreased significantly in 24 h [23]. NZnO has been added to resin or binder alone or in cooperation with other antibacterial agents (Table 1). As recent researches asserted the antimicrobial effects of NZnO and chitosan (CS) nanoparticles, which results demonstrated that the two components of nanoparticles in the composite resin can be synergistically beneficial in reducing the number of microorganisms [24]. Yazi Wang et al. explored the mechanical and antibacterial properties of cellulose nanocrystal/zinc oxide (CNC/ZnO) nanohybrids of the dental resin composites [25]. They concluded that CNC/ZnO nanohybrids positively affected the mechanical and antibacterial properties on dental composite resin which are promising to resist secondary caries. In the dentine caries model, Mario et al., found that NZnO and copper nanoparticles (NCuO) addition in universal adhesive systems provided antimicrobial, anti-MMP activities and improved interface stability on caries-affected dentin [26].

### 2.3. Other Metal Nanoparticles

TiO_2_ nanoparticles (NTiO_2_) were incorporated into glass-ionomer at 3% and 5% (*w*/*w*), exhibiting an antibacterial activity in direct contact test against *Streptococcus mutans* [27,28]. Moreover, *N*TiO_2_-containing dental adhesives (80% *v*/*v*) had strong antibacterial efficacy against *Streptococcus mutans* biofilms [29] with good biocompatibility [30]. In vitro experiment, an acrylic resin containing NTiO_2_ (0.5%) and nano-SiO_2_ (NSiO_2_, 1%) in acrylic liquid reduced the cariogenic bacterial count (*Lactobacillus acidophilus and Streptococcus mutans*) by 3.2–99% in a time-dependent manner [31]. Toodehzaeim MH et al. indicated that incorporating NCuO into adhesive (0.01, 0.5, and 1 wt.%) added antimicrobial effects to the adhesive with no adverse effects on shear bond strength [32]. Besides, the incorporation of Cu nanoparticles into adhesive renders the adhesive antibacterial to *Streptococcus mutans* for at least 1 year [33,34]. The MgO nanoparticles (NMgO) modified glass-ionomer cement showed effective antibacterial and antibiofilm activity against two cariogenic microorganisms (*Streptococcus mutans* and *Streptococcus sobrinus*) [35].

### 2.4. The Anti-Caries Mechanism of Mental Nanoparticles

The antibacterial mechanism of mental nanoparticles is summarized in Figure 1.

The first is “Contact Inhibited Mechanism”. For metal nanoparticles, the positively charged metal ions contact with negatively charged cell membranes on microorganisms, which is called the micro-dynamic effect. Metal ions penetrate the cell membrane and enter the microbial body. They react with the thiol group (-SH) on the microbial protein, inhibiting the synthesis of protein and nucleic acid. Metal ions destroy the electron transport system, respiratory system, and material transfer system of microorganisms, resulting in microbial death. Metal ions are generally loaded on sustained-release carriers and gradually released in the reaction process. When the bacteria were killed by metal ions (such as Ag^+^), the metal ions are free from the bacterial corpse and can contact with other bacteria again. In this way, metal ions can maintain a durable antibacterial effect. The bactericidal and inhibitory activities of metal ions decreased in the following order: Ag^+^ > Hg^2+^ > Cu^2+^ > Cd^2+^ > Cr^3+^ > Ni^2+^ > Pb^2+^ > Co^3+^ > Zn^2+^ > Fe^3+^.

The second is “Reactive Oxygen Species (ROS) Mechanism”. According to the hypothesis of the ROS mechanism, the trace metal elements distributed on the surface of the materials are the active centers that play a catalytic role. The active center can absorb energy in the environment, activating the oxygen on the material surface, and generating hydroxyl radicals and reactive oxygen atoms. Their strong redox effect would destroy the cell proliferation ability to inhibit or kill bacteria [36]. Yoshihiro [37] et al. showed that only under aerobic conditions can silver loaded zeolite to exhibit antibacterial activity, which provided strong evidence for the mechanism of ROS.

## 3. Quaternary Ammonium Salt Polyethylenimine (QAS-PEI) Nanoparticles

QAS is a highly active cationic agent with a wide antibacterial spectrum [38,39]. QAS-PEI nanoparticles were prepared based on polyethyleneimine cross-linked structure which makes the modified composite with high chemical stability and antibacterial properties under different oxidants and storage conditions, without effect on the oral micro ecological balance. The antibacterial mechanism of QAS-PEI is related to the electrostatic interaction between positively charged QAS-PEI and negatively charged bacterial cell walls [40]. QAS-PEI nanoparticles incorporated in resin-based composite at 2% *wt*/*wt* have demonstrated prolonged inhibition of bacterial growth [41]. Yudovin Farber et al. [42] added 1% QAS-PEI nanoparticles into the composite resin. Through 3 months of long-term experiments in vitro, it was confirmed that the composite resin had strong sustained antibacterial activity against *Streptococcus mutans*, and the alkyl chain length of polyethyleneimine had a significant influence on the antibacterial effect. In addition, some scholars have added QAS-PEI nanoparticles into glass ionomer polymer and found it has an obvious antibacterial effect on *Streptococcus mutans* and *Lactobacillus* [43]. Moreover, Nurit Beyth et al. investigated the biocompatibility of QAS-PEI nanoparticles incorporated in a resin composite [30,44], revealing that TNFα secretion and macrophage viability were not altered by QAS-PEI nanoparticles.

## 4. Remineralized Nano Anti-Caries Materials

Remineralized nano anti-caries materials are adhesives or resins that added with bioactive nanoparticles which can remineralize the exposed collagen in the mixed layer and prevent collagen degradation, improving the bonding durability. Remineralized nanoparticles modified resins can release Ca^2+^ and PO4^3−^ in low pH values and promote the remineralization of tooth tissue to prevent caries.

### 4.1. Nano Particulate Hydroxyapatite (NHAP)

The physical and chemical properties of the synthesized NHAP are similar to those of natural tooth hydroxyapatite. NHAP can stably release Ca^2+^ and PO4^3−^ to promote remineralization. Vyavhare et al. [45] found that NHAP can precipitate on the surface of demineralized enamel to form a new layer. NHAP enhanced the shear bond strength to remineralized enamel in the etched enamel model [46]. Moreover, in the recurrent enamel caries model, resin infiltration doped with NHAP caused higher enamel resistance against demineralization compared to the control group [47]. Roza Haghgoo et al. studied the efficacy of different NHAP concentrations in mouthwash for the remineralization of incipient caries [48]. The results showed that NHAP can significantly enhance tooth remineralization and increase microhardness in a dose-dependence manner. NHAP added in total etching adhesive can increase the micro shear bond strength of teeth [49]. However, it will weaken the mechanical interlock between the adhesive and the tooth when added in the self-etching adhesive. In vivo and in vitro, toothpaste containing NHAP has proven to be a valuable prevention measure against dental caries in primary dentition [50].

### 4.2. Nanosized Calcium Fluoride (NCaF_2_)

Compared with the traditional fluoride, NCaF_2_ materials can keep fluorine release at a better level for a long time. NCaF_2_ can be added to dental composites as dental filling compo glass with a size of 58 ± 21 nm [51]. The results showed that the fluorine release concentration in NCaF_2_-composite resin is similar to or even higher than that of commercial modified glass ionomer. Moreover, after thermal cycling and 2-year aging treatment, the flexural strength of NCaF_2_-composite resin is 5 times higher than that of commercial modified glass ionomer [52]. However, the aesthetic properties still need to be improved due to the mismatching of the refractive index between filler and matrix. In a white spot lesion model, NCaF_2_-containing orthodontic cement increased the enamel hardness by 56% and decreased the lesion depth by 43%, compared to control groups [53]. In caries prevention, a novel pit and fissure sealant containing NCaF_2_ and antibacterial component achieved high F release and strong antibacterial performance [54]. In a rat caries model, NCaF_2_ was observed to substantially decrease caries by scanning electron micrographs [55]. Thus, NCaF_2_ materials is promising to inhibit enamel demineralization, white spot lesions, and caries.

### 4.3. Nanosized Amorphous Calciumphosphate Particle (NACP)

Compared with amorphous calcium phosphate, NACP can release higher levels of Ca^2+^ and PO4^3−^ with lower filler. Lee et al. [56] found that NACP can release more ions at low pH, with the acid invasion neutralization, increasing the pH value from 4 to 6.5 to resist dental caries. Besides, studies have shown that the addition of NACP in adhesive can improve the pulp biological reaction and promote tertiary dentin formation [57]. When NACP combined with antibacterial components are added to the adhesive, the new binder has dual functions of both antibacterial and remineralization [58]. Novel adhesive containing MBDP and NACP exhibited an application prospect in resisting enamel white spot lesions and even caries [59]. Although NACP composite resin has strong remineralization ability, Weir et al. [60] revealed that the ion release of NACP can only last for a few months. To solve this problem, Zhang et al. [61,62] developed a composite resin which can re-release Ca^2+^ and PO4^3−^. The new resin has the function of long-term anti-caries and auxiliary remineralization, in which mechanical properties have no significant difference with commercial composite materials. Moreover, Yousif A. Al Dulaijan et al. investigated a novel rechargeable calcium phosphate synthesized with NACP and dimethylaminohexadecyl methacrylate (DMAHDM), which can recharge and re-release Ca and P ion [61]. Moreover, this release remains at the same level as the charge cycle increases, indicating the long-term ability of ion release and remineralization. In the artificial enamel caries model, an adhesive containing NACP could achieve higher enamel remineralization of artificial caries than the control by surface and cross-sectional hardness test in vitro [59,63,64]. In the artificial dentin caries model, the NACP adhesive remineralized dentin lesions with exceptional efficacy [65,66] and showed excellent water-aging durability for 12 months [67]. In a biofilm-based recurrent root caries model, NACP-containing nanocomposite substantially reduced demineralization and protected root dentin hardness around the restorations [68]. In a rat tooth cavity model, the antibacterial and remineralizing properties of adhesive containing NACP and dimethylaminododecyl methacrylate (DMADDM) were tested in vivo. The novel adhesive exhibited much greater tertiary dentin formation and milder pulpal inflammation than the control [57]. Besides, NACP combined into an antibacterial sealant protected the enamel against demineralization in vivo [69]. Thus, pH-responsive Ca^2+^ and PO4^3−^ releasing sealants may be a reliable complementary approach for caries management [69,70].

### 4.4. Bioactive Glass Nanoparticle (NBG)

When the bioactive glass contacts with water or saliva, it will release Ca^2+^ and PO4^3−^ to form a mineralized layer with a porous network, which is similar to hydroxyapatite. NBG has better physical and chemical activities due to the decrease of grain size, the increase of surface area, the surface free energy, and the binding energy. The research showed that NBG can interfere with the degradation of collagenase and maintain a high alkaline pH, resulting in antibacterial ions (such as Ag^+^) release to achieve antibacterial effect [71]. Camila et al. prepared nanocomposite cements based on the incorporation of NBGs into BiodentineTM and to assess their bioactive properties. They concluded that the incorporation of NBGs into BiodentineTM improved in vitro bioactivity of modified resin, accelerating the formation of a crystalline apatite layer on the resin surface [72]. Tauböck et al. [73] demonstrated that the composite resin containing 20% NBG formed a mineralized layer after 21 days, while the composite resin without NBG had slight corrosion and no sediment formation. However, it needs further study that whether the increase of BGN content would affect the hardness of the composite resin since its hydrophilicity and water absorption rate were increased.

### 4.5. Remineralization Mechanism of Nanoparticles

Caries demineralization involves loss of minerals occurring in the early stage of the lesion, in the deep below the enamel surface, with the migration of acid ions from the plaque to the deep lesion and mineral ions from the deep lesion toward the plaque [74]. Caries remineralization is a natural repair process to restore the minerals in ionic forms to achieve the hydroxyapatite crystal lattice [75,76]. It occurs at near-neutral physiological pH, and calcium and phosphate mineral ions in nanoparticles are redeposited from saliva and dental plaque fluid to form new hydroxyapatite crystals, which are larger and more acid-soluble.

## 5. Nanodrug Delivery System

Nano drug delivery system is a new drug controlled-release system with a particle size of 1–100 nm, which is made of carrier materials and anti-caries drugs by nanotechnology. The anti-caries effect can be enhanced by adding a nanodrug loading system into composite resin or adhesive. In recent years, the system showed a promising application in the clinic with the advantages of targeting and sustained drug release.

### 5.1. Mesoporous Silica Nanoparticle (MSN)

The surface of MSN can be modified by functional groups, which made MSN can be compatible with various solutions and stored as different types of molecules. Besides, due to the high affinity of MSN, it is easy to adhere to the dentin surface. Calcium and phosphate can be slowly released from MSN, which can improve the bonding durability and remineralization effect [77,78]. The study of chlorhexidine on MSN showed that MSN had a high inhibitory ability on a variety of oral bacterioplankton and biofilm [79]. Zhang et al. [80] found that compared with the composite resin containing chlorhexidine only, MSN loaded chlorhexidine composite resin has the characteristics of slow-release, longer antibacterial effect, and relatively stable flexural strength. Moreover, Cameron A. et al. have developed new broad-spectrum antimicrobial agent MSNs co-assembled with the template drug, octytening salt. They found that the steady-state release of template agent in adhesive killed caries-related bacteria, inhibiting biofilm formation on the adhesive surface and non-toxic [81].

### 5.2. Liposomes

A liposome is a microvesicle formed by encapsulating drugs in the lipid bilayer. Gregoriadis et al. [82] used liposomes as drug carriers for the first time. Liposomes can encapsulate lipophilic or water-soluble drugs which have been widely studied as anti-cancer, antibacterial, anti-inflammatory drug carriers as well as for gene delivery due to its advantages of targeting, slow-released and tissue affinity after encapsulating drugs. In the caries prevention and treatment, liposome drug delivery can induce calcium, phosphorus, and other minerals to precipitate on the surface of tooth hard tissue which can promote mineralization. Fat-soluble or water-soluble antimicrobial agents encapsulated in liposomes have a good effect on plaque biofilms. The latest study [83] found that in addition to the role of drugs carried by liposomes, a new type of pectin coated liposome carrier can permanently adsorb on the enamel surface to protect the enamel.

### 5.3. Halloysite Nano-Tube (HNT)

Halloysite is a kind of natural silicate mineral, which mainly exists in nature in the form of nanotubes. HNT can be used as a carrier to control the delivery of therapeutic agents with excellent biocompatibility, hydrophilicity, and high mechanical strength [10]. NZnO and NAg were loaded into HNT by some scholars. It was found that HNT promoted the dispersion of NZnO, made ZnO contact closely with bacteria and increased the concentration of zinc, which achieved a synergistic antibacterial effect with NAg [84]. Bottino et al. [85] found that HNT can be used as a carrier of MMP inhibitor in a modified total-etching adhesive to increase the adhesion durability, and the addition amount can be up to 20%, without affecting the physical and chemical properties. S.A. Feitosa et al. investigated the biological activity and bonding properties of doxycycline (DOX)-encapsulated HNT-modified adhesive. The results showed the growth of *Streptococcus mutans* was successfully inhibited by direct contact. In addition, compared with the control group, the DOX-encapsulated HNT group had stronger inhibitory activity on MMP-1 [86].

### 5.4. Polyamidoamine (PAMAM)

A dendrimer is a novel type of high branched nano polymer with a cavity in the molecule and easy to modify surface groups. It has good biocompatibility, low toxicity, and non-immunogenicity. PAMAM terminal groups can be modified into various functional groups as drug delivery carriers to control drug release and play dual functions of both antibacterial and remineralization in caries. Zhou et al. [87] found that PAMAM-COOH can be used as a carrier of triclosan to improve the antibacterial effect of triclosan and induce remineralization. Moreover, a bovine enamel carious lesions model was created to investigate the remineralization ability of PAMAM with different terminal groups, among which PAMAM-NH_2_ showed the most prominent competence, followed by PAMAM-COOH and PAMAM-OH, in that order. In the dentin caries model, the PAMAM combined with NACP adhesive completely remineralized demineralized dentin for long-term fluid challenges to protect tooth structures [88,89]. In a biofilm-based recurrent root caries model, the combined bioactive multifunctional composite (BMC) with PAMAM substantially induced root dentin remineralization and increased the hardness of pre-demineralized root dentin [90]. Liang et al. [91] studied the effects of PAMAM, amorphous nano calcium phosphate, and their mixed adhesives on remineralization. The result showed that compared with single-component adhesives, mixed adhesives can promote dentin remineralization more effectively in an acidic environment. Researchers loaded amphotericin B into the MSN-incorporated PMMA and observed a long-term antimicrobial effect for two weeks, which suggested that MSN-incorporated PMMA resin had clinical application prospect [92].

### 5.5. Dental Caries Vaccine

It is an effective strategy in caries prevention to induce oral mucosal immune systems through the nasal tract. Vaccination has the following advantages: High patient compliance, induction of systemic immunity, and convenient administration [93,94]. Recently, promising results have been achieved in the study of anti-caries DNA vaccine inoculation [95]. However, due to the lack of effective targeting, the effectiveness of the mucosal administration of “naked” DNA is limited. In order to improve the function of the “naked” DNA vaccine, a plasmid was loaded into chitosan nanoparticles [96]. Chitosan is a rich natural biopolymer extracted from the exoskeleton of crustaceans [97]. In animal experiments, chitosan has been proved to be an excellent vaccine carrier system [96] and a vehicle suitable for oral drug delivery systems [98]. Chitosan is a valuable gene carrier in oral [99]. The chitosan-DNA nanoparticles are suitable for mucosal anti-caries DNA vaccination, which could induce the oral specific immune responses with biocompatible and non-cytotoxic [100]. In further research, human clinical trials are needed to implement regular anti-caries strategies.

## 6. Biomimetic Nanocatalyst

The application of nanocatalyst is a new approach to combat cariogenic plaque-biofilm in nanotechnology. In a previous study, catalytic iron oxide nanoparticles (CAT-NP) have been shown exceptional topical anti-biofilm effects in vitro [101]. Moreover, CAT-NP/H_2_O_2_ was observed to suppress the onset and severity of dental caries in vivo in a rodent model [101]. With an in-depth study, Pratap C. Naha et al. reported that dextran-coated iron oxide nanoparticles termed nanozymes (Dex-NZM) displayed strong catalytic activity at acidic pH values and targeted biofilms with high specificity, preventing the development of dental caries in a pH-dependent manner in vivo [102].

## 7. Conclusions

The anti-caries components of nanoparticles were summarized in Table 1. The roles of anti-caries nanomaterials were exhibited in Figure 2. With the development of caries diagnosis and prevention, nanotechnology will significantly improve medical treatment. The use of nanoparticles against caries infection is very important because of its antibacterial, remineralization, and drug loading capacity. All the studies in this review have shown that nanoparticles have an enhanced role in the prevention and treatment of caries infection. However, it is necessary to understand their disadvantages and their potential cytotoxicity and environmental effects. In the future, we should develop better technology to prepare highly effective anti-caries nanoparticles with the highest safety for patients.

**Table 1 molecules-25-05047-t001:** The anti-caries component of nanoparticles.

Type	Component	Modified Materials	Concentration	Model In Vitro Experiment	Mechanism of Anti-Caries	Ref.
**Metal Nanoparticles**	**Silver Nanoparticle (NAg)**	Resin Adhesive	Resin: 0%~0.088% Adhesive: 1–5%	*Streptococcus mutans* and *Lactobacillus*	①Ag ions penetrate the cell membrane and enter the microbial body. ②Mechanism of reactive oxygen species (ROS).	[103]
	**NAg-NZnO**	Resin	NAg, 1% NZnO, 1%	*Streptococcus mutans* and *Lactobacillus*		[104]
	**NAg-Laden Hydroxyapatite**	Resin	6–8 wt.%	*Streptococcus mutans*	①Antibacterial mechanism of NAg. ②Hydroxyapatite can precipitate on the surface of demineralized enamel to form a new layer and promote remineralization.	[105][45]
	**NAg- Polyamidoamine (PAMAM)-Cellulose Nanocrystals (CNCs)**	Resin	0.1 wt.%	*Staphylococcus aureus* and *Escherichia coli*	Antibacterial mechanism of NAg and remineralized mechanism of PAMAM.	[14]
	**NAg-Quaternary Ammonium Dimethacrylate (QADM)**	Adhesive	NAg, 0.05% QADM, 10%	*Streptococcus mutans*	①Antibacterial mechanism of NAg. ②Mechanism of quaternary ammonium salt QAS. *CHK1*-mediated two-component regulatory system results in the accumulation of ROS which induces cell apoptosis.	[106]
	**NAg- 12-methacryloxydodecylpyridium bromide (MDPB)**	Adhesive	NAg, 0.1% MDPB, 2.5%	Human saliva biofilms		[107]
	**NAg- Dimethylaminododecyl methacrylate (DMADDM)**	Adhesive	NAg, 0.1% DMADDM, 5%	Dental plaque microcosm biofilm model		[108]
	**Nano-Zinc (NZn)**	Adhesive	2.15 ± 0.05 µg Zn/mg NPs	Extracted unerupted human third molars	Inhibit MMP activity, reduction of the decomposition of dentin collagen bundle, protection of mineral crystal at the interface of resin-tooth formation.	[109]
	**Nano-Zincoxide (NZnO)**	Resin	1%	*Streptococcus mutans*	①NZnO has higher surface potential energy and can release more zinc ions to kill bacteria. ②NZnO can also activate the photocatalytic antibacterial mechanism and produce a large number of free radicals to interact with bacteria.	[21]
	**TiO_2_ nanoparticles (NTiO_2_)**	Glass-ionomer	3% and 5%	*Streptococcus mutans*	①Contact inhibited mechanism. ②Mechanism of reactive oxygen species (ROS).	[28]
	**Nano-SiO_2_ (NSiO_2_)**	Acrylic resin	1%	*Lactobacillus acidophilus* and *Streptococcus mutans*		[31]
	**Copper Nanoparticles (NCuO)**	Adhesive	0.01, 0.5, and 1 wt.%	*Streptococcus mutans*		[32]
	**MgO Nanoparticles (NMgO)**	Glass-ionomer	1% and 2.5%	*Streptococcus mutans* and *Streptococcus sobrinus*		[35]
**QAS**	**Quaternary Ammonium Salt Polyethylenimine (QAS-PEI) Nanoparticles**	Resin	1–2 wt.%	*Streptococcus mutans and Lactobacillus*	The electrostatic interaction between positively charged QAS-PEI and negatively charged bacterial cell walls.	[43]
**Remineralized Nanopaticles**	**Nano Particulate Hydroxyapatite (NHAP)**	Resin	2–5–10%	Sound premolars fixed in acrylic blocks and coated with nail polish	NHAP can stably release Ca^2+^ and PO4^3−^ to promote remineralization.	[48]
	**Nanosized Calcium Fluoride (NCaF_2_)**	Resin	17%	Biofilm by *Streptococcus mutans* on the tooth surface	NCaF_2_ materials can keep fluorine release at a better level for a long time to promote tooth remineralization.	[51]
	**Nanosized Amorphous Calciumphosphate Particle (NACP)**	Resin	Ca and P with concentrations of 8 mmol/L and 5.333 mmol/L	Dental plaque microcosm biofilm model	NACP can release higher levels of Ca^2+^ and PO4^3−^ at low pH, with the acid invasion neutralization, increasing the pH value from 4 to 6.5 to resist dental caries.	[61,62]
	**Bioactive Glass Nanoparticle (NBG)**	Resin	20 wt%	Bioglass	①It will release Ca^2+^ and PO4^3−^ to form a mineralized layer with a porous network. ②NBG can interfere with the degradation of collagenase, formation of high alkaline pH, resulting in antibacterial ions (such as Ag^+^) release to achieve an antibacterial effect.	[73]
**Drug Delivery System**	**Mesoporous Silica Nanoparticle (MSN)**	Adhesive	34 wt%	Multi-species biofilms	As a carrier, the system can slowly release antibacterial/remineralization particles.	[81]
	**Liposome**	-	0.05%, 0.2%	Dental enamel, Saliva		[83]
	**Halloysite Nano-Tube (HNT)**	Adhesive	20%	*Streptococcus mutans*		[86]
	**PAMAM**	Adhesive/resin	0.3% (*w*/*v*)	Dentin disks, Artificial saliva		[87]
	**Caries Vaccine**	-	50 mmol/L	Plasmid		[100]
**Nanocatalyst**	**Catalytic iIon Oxide Nanoparticles (CAT-NP)**	gargle	4%	a rodent caries model	As a catalyst, it can catalyze the effect of H_2_O_2_ against cariogenic bacteria.	[101]
	**Dextran-Coated Iron Oxide Nanoparticles termed Nanozymes (Dex-NZM)**	gargle	-	a rodent caries model	As a catalyst, it can catalyze the effect of Fe_4_O_3_ against cariogenic bacteria.	[102]

## Figures and Tables

**Figure 1 molecules-25-05047-f001:**
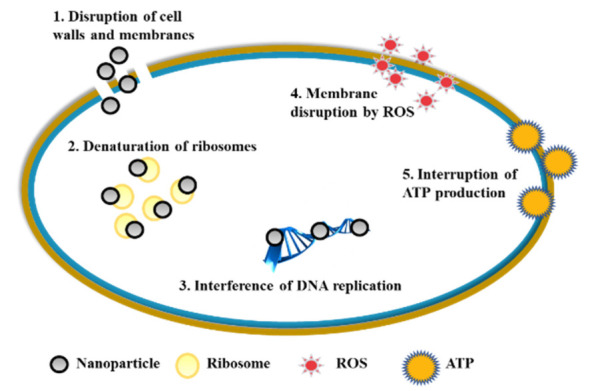
The antibacterial mechanism of mental nanoparticles.

**Figure 2 molecules-25-05047-f002:**
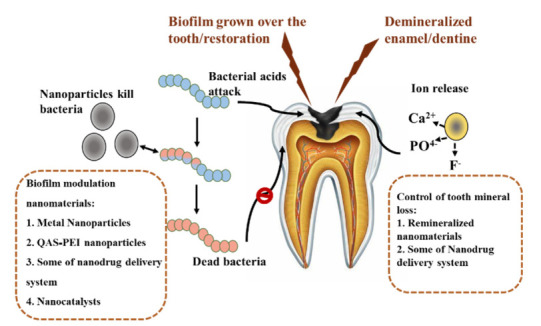
Schematic is indicating the roles of anti-caries nanomaterials.

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
