# Peer review of "Advances of Anti-Caries Nanomaterials"

_molecules, 2020, doi:10.3390/molecules25215047_

Round 1
Reviewer 1 Report
In this narrative review, authors presented "Advances of anti-caries nanomaterials".
However, authors presented many conventional nano-materials e.g. silver, zinc oxide, calcium phosphate. So the word "advances" should be justified with the addition of other nanomaterials which have been used in dental restorative materials and has shown some anti-caries effect e.g. gold nanoparticles, copper, magnesium alone or in combination with bioceramics.
General Comments:
Authors should focus on typo errors and need to be corrected.
Repetition of some references. Carefully read all references.
Authors should support their statement about the roles of nanomaterials anti-caries with images/figures.
Authors should concentrate more on in vivo effect and in in vitro, focus should be more on effect of these materials on carious dentine, carious enamel, white spot lesion etc. Support this with Figures.
It should be clearly mentioned that either these nanomaterials have been used in resin-based composites (for filling purpose) or in adhesives/bonding agents. Re-arrange statements according to this.
Author Response
Response to reviewer’s comments:
1. Comments: Authors should focus on typo errors and need to be corrected.
Response: Thank you so much. We have corrected typo and grammatical errors, which are indicated in red font.
2. Comments: Repetition of some references. Carefully read all references.
Response: Thank you very much. We have removed repetitive references and reordered them.
3. Comments: Authors should support their statement about the roles of nanomaterials anti-caries with images/figures.
Response: We gratefully appreciate for your comment. We added Fig. 2, in line 289, which indicated the roles of anti-caries nanomaterials.
4. Comments: Authors should concentrate more on in vivo effect and in in vitro, focus should be more on effect of these materials on carious dentine, carious enamel, white spot lesion etc. Support this with Figures.
Response: Thank you for your valuable advice. Fig.3 (in line 292) summarized the type of experiments in nanomaterials.
5. Comments: It should be clearly mentioned that either these nanomaterials have been used in resin-based composites (for filling purpose) or in adhesives/bonding agents. Re-arrange statements according to this.
Response: We gratefully appreciate for your valuable suggestion. At present, these anti-caries nano components in resin/adhesives are only carried out in vitro experiments. In Table 1, we showed the addition of various anti-caries nano components in resin/adhesives.
Reviewer 2 Report
The review article by Chen et al is a very interesting and well written update of the current state of the art on Advances of anti-caries nanomaterials.
Indeed, the review may catch the interest of not only medical dentists but also materials scientists, chemists and biologists working in develop future strategies fighting teeth infection. In fact, reading this article and some of its references a new idea has prompted into my mind to develop anticaries dental implants. I did not identify significant problems on the review.
In my opinion the chapters of the article are appropriate, and the discussion shows the pros and cons of using nanomaterials and potential limitations.
There are minor points to be revised:
I missed some comments on the market available using nanoparticles.
It is not clear if any or the materials described has been moved to in vivo applications or already commercialized. If so, it will be interesting to know why or why not a particular material can be produced industrially and placed in real clinic.
I recommend the publication of this review article after minor revision.
Author Response
Response to reviewer’s comments:
Comments: I missed some comments on the market available using nanoparticles.
It is not clear if any or the materials described has been moved to in vivo applications or already commercialized. If so, it will be interesting to know why or why not a particular material can be produced industrially and placed in real clinic.
Response: We gratefully thanks for your constructive suggestions. At present, these anti-caries nano components in resin/adhesives are only carried out in vitro experiments. Fig.3 (in line 292) summarized the type of experiments in nanomaterials. Despite the rapid development of nano dental technology, safety and cost are still obstacles for its entry into the market. Since the nanomaterials may be toxic to human cells, formal clinical trials are required before any new nanomaterials can be used in dental treatment. These nanomaterials may be more expensive, and insurance companies may refuse to pay for the nanomaterials treatment if they are identified as cosmetic items. However, the researchers believe that these obstacles can be overcome in the future, and the new nano materials will be widely used in a few years.
Round 2
Reviewer 1 Report
Comments:
Many typo errors. Authors should proofread it.
It was suggested to add other nanomaterials which have been used in dental restorative materials and has shown some anti-caries effect e.g. gold nanoparticles, copper, magnesium alone or in combination with bioceramics. Authors should incorporate those.
It was suggested to add information about in vitro and in vivo experimental work. Authors added a statement in Conclusion and supported with Figure (which is not sufficient). Authors should incorporate experimental procedures (in vitro and in vivo at appropriate places. Many experiments have been done on enamel and dentine. It should be described clearly in this narrative review.
Author Response
Response to reviewer’s comments:
1. Comments: Many typo errors. Authors should proofread it.
Response: Thank you so much. We have corrected typo and grammatical errors, which are indicated in deep red font.
2. Comments: It was suggested to add other nanomaterials which have been used in dental restorative materials and has shown some anti-caries effect e.g. gold nanoparticles, copper, magnesium alone or in combination with bioceramics. Authors should incorporate those.
Response: We gratefully appreciate your comment. We added other nanomaterials used in dental materials in red font (in lines 114-126, 329-336). Please check our new version.
3. Comments: It was suggested to add information about in vitro and in vivo experimental work. Authors added a statement in Conclusion and supported with Figure (which is not sufficient). Authors should incorporate experimental procedures (in vitro and in vivo at appropriate places. Many experiments have been done on enamel and dentine. It should be described clearly in this narrative review.
Response: Thank you for your valuable advice. We added information about in vitro and in vivo experimental work in red font (in lines 83-89, 111-114, 177-179, 185-186, 195-199, 218-229, 302-309). Please check our new version.
